# Controlled Recirculating Wet Storage Purging *V. parahaemolyticus* in Oysters

**DOI:** 10.3390/pathogens11050553

**Published:** 2022-05-07

**Authors:** Ruojun Mu, Chengchu Liu, Salina Parveen, Donald Webster, Jie Pang

**Affiliations:** 1College of Food Science, Fujian Agriculture and Forestry University, Fuzhou 350002, China; 000q819034@fafu.edu.cn; 2Sea Grant College Extension Program, University of Maryland Extension and Maryland-Sea Grant Extension, Princess Anne, MD 21853, USA; 3Department of Agriculture, Food and Resource Sciences, University of Maryland Eastern Shore, Princess Anne, MD 21853, USA; sparveen@umes.edu; 4Wye Research & Education Center, University of Maryland Extension, Queenstown, MD 21658, USA; dwebster@umd.edu

**Keywords:** eastern oysters (*Crassostrea virginica*), depuration, wet-storage, *V. parahaemolyticus*, seafood safety

## Abstract

This work explored the effects of salinity and temperature on the efficacy of purging *V. parahaemolyticus* from eastern oysters (*Crassostrea virginica*). Oysters were inoculated with a 5-strain cocktail of *V. parahaemolyticus* to levels of 10^4^ to 10^5^ MPN (most probable number)/g and depurated in a controlled re-circulating wet-storage system with artificial seawater (ASW). Both salinity and temperature remarkably affected the efficacy for the depuration of *V. parahaemolyticus* from oysters during wet-storage. The wet-storage process at salinity 20 ppt at 7.5 °C or 10 °C could achieve a larger than 3 log (MPN/g) reduction of *Vibrio* at Day 7, which meets the FDA’s requirement as a post-harvest process for *V. parahaemolyticus* control. At the conditions of 10 °C and 20 ppt, a pre-chilled system could achieve a 3.54 log (MPN/g) reduction of *Vibrio* in oysters on Day 7. There was no significant difference in the shelf life between inoculated and untreated oysters before the depuration, with a same survival rate (stored in a 4 °C cooler for 15 days) of 93%.

## 1. Introduction

*V. parahaemolyticus* is a human pathogen that naturally exists in marine environments [1] and primarily causes seafood-associated gastroenteritis in the U.S., mostly from the consumption of raw oysters [2]. Based on CDC data, *Vibrio* infections follow a seasonal trend mainly from June to September at high temperatures [3]. Oysters can uptake and accumulate *Vibrio* from growth environments [4]. *Vibrio* can also replicate in oysters at favorable temperatures [5]. Therefore, a naturally higher abundance of *V. parahaemolyticus* (<10,000 [most probable number] MPN/g) can be detected in oysters [6]. It is reported that around 20 million Americans consume raw shellfish [7]. It has been documented that many outbreaks of *V. parahaemolyticus* infections were related to eating raw oysters in the U.S. since 1973 [8,9].

To increase shellfish safety, the seafood industry employs different strategies intended to reduce or control pathogen levels in oysters [10]. Liu et al. [11] revealed that frozen storage could achieve around 3.52 log (MPN/g) reductions of *V. parahaemolyticus* in half-shell Pacific oysters. Ma et al. [12] identified and validated a high hydrostatic pressure processing (HPP) method for commercial applications that achieved greater than 3.52-log reductions of *V. parahaemolyticus* in Pacific oysters. However, both methods lead to the death of the oysters; reducing the *Vibrio* levels in live oysters is still a challenge. Depuration in UV-sterilized water and electrolyzed oxidizing water was used to solve this problem [13,14]. Irradiations can also reduce the levels of *Vibrio* while keeping oysters alive [15]. One previous study reported that a wet-storage process (7–15 °C for 5–7 days) could reduce lab-inoculated *V. parahaemolyticus* in Pacific oysters (Crassostrea gigas) greater than 3.0 log units [16]. However, no information is available on the wet-storage process that is applied in reducing *V. parahaemolyticus* in eastern oysters (*Crassostrea virginica*).

In this work, we investigated the effects of salinity and temperature on the efficacy of purging *V. parahaemolyticus* from eastern oysters (*Crassostrea virginica*). We inoculated oysters with *V. parahaemolyticus* to levels of 10^4^ to 10^5^ MPN/g and depurated them in a controlled re-circulating wet-storage system with artificial seawater (ASW). We explored the effects of salinity, temperature, and the system design on the reduction of the *Vibrio*, as well as the shelf life of the oysters. In this study, we focused on the reduction of *V. parahaemolyticus* in eastern oysters and applied a new design of the wet-storage system. We hope those will give new insight into the depuration of *Vibrio* in seafood.

## 2. Results

### 2.1. Occurrence and Accumulation of V. parahaemolyticus in Oyster

The occurrence of *V. parahaemolyticus* levels in oysters was significantly different based on the date of harvest (Table 1). Winter- and early spring-harvested oysters had extremely low levels of *V. parahaemolyticus* levels (<1 log MPN/g), while summer-harvested oysters showed a significant increase in the occurrence of *V. parahaemolyticus* levels (about 3 log MPN/g). Based on the results, the inoculation procedure can accumulate *V. parahaemolyticus* levels to ~5 log MPN/g. Most importantly, this inoculation procedure is not dependent on the background level of *V. parahaemolyticus* in oysters.

### 2.2. Effects of Salinities on the Reduction of Vibrio

We firstly explored the effect of salinity on the reduction of *Vibrio*, controlling the temperature at 10 °C (Table 2). In all three salinity conditions, levels of *V. parahaemolyticus* were significantly different during the first 2 days, reduced up to 1 log MPN/g (salinity 15 ppt, Day 2). From Day 3, the reduction slowed down to around 0.3 log MPN/g per day. Those results revealed that our wet-storage system had a high deficiency during the first 2 days of the depuration. Depuration of oysters in ASW at 20 ppt resulted in 3.14 log MPN/g of reduction of *V. parahaemolyticus* in oysters after 7 days, while at 15 and 25 ppt ASW, the reduction only achieved 2.85 and 2.77, respectively.

### 2.3. Effects of Temperatures on the Reduction of Vibrio

We further explored the effect of temperature on the reduction of *Vibrio* in 20 ppt ASW, which we had already confirmed is the best salinity for depuration. We chose 3 temperatures to test the changes in the *V. parahaemolyticus* levels during the 7-day wet-storage process (Table 3). With the temperature decreased, the reduction of the *V. parahaemolyticus* levels significantly increased. At a temperature of 7.5 °C, the log reduction of the *Vibrio* achieved a 3.38 log MPN/g, which was the highest value among all trials. The significant increase in the depuration efficiency of the wet-storage was attributed to the low temperature. The levels of *V. parahaemolyticus* significantly decreased during the first 4 days, which was 2 days longer than all other trials. Detailed reductions (Log_10_ MPN/g) of *V. parahaemolyticus* at 7.5 °C, the salinity of 20 ppt is shown in Figure 1.

### 2.4. Effects of System Pre-Chilling on the Reduction of Vibrio

To get better efficiency of depuration for our wet-storage system, we pre-chilled the water tank before the wet-storage process. Figure 2 illustrates the temperature reduction rate of the wet-storage system. It took about 5 h for the system to decrease the temperature to 10 °C. Figure 3 shows the reductions of *V. parahaemolyticus* in inoculated oysters during the pre-chilled (black bar) and non-pre-chilled (grey bar) wet-storage processes. The temperature and salinity of the ASW were controlled at 10 °C and 20 ppt. The reduction of the *V. parahaemolyticus* levels in a pre-chilled wet-storage process significantly increased to 3.54 log MPN/g. This value is 0.82 log MPN/g higher than a non-pre-chilled depuration system. Furthermore, the pre-chill significantly increased the reduction of the *V. parahaemolyticus* levels on each day of depuration and the reduction rate.

### 2.5. Shelf Life of the Oyster

A total of three studies (100 oysters per study) were conducted to evaluate the survival rate of oysters in the walk-in cooler after the accumulated treatments (Figure 4). All oysters survived 3 days of refrigerated storage. For oysters after inoculation and without any treatment, only one oyster died on Day 5. All accumulated and non-treated oysters had similar survival rates during 15 days of storage at 4 °C. The survival rate of inoculated, and untreated oysters before depuration are both 93%. These results indicate that accumulated treatments do not affect the shelf life of oysters for depuration.

## 3. Discussion

To prepare a unified level of *Vibrio* in original oysters for depuration, we scheduled the inoculation method for the accumulation of *Vibrio* in oysters. Even when we could detect naturally occurring *V. parahaemolyticus* in oysters harvested in summer, we still needed inoculation to increase the level to specific levels for our wet-storage depuration test. Our results are mostly consistent with findings from current studies [1,2,17]. Decreases in the level of *V. parahaemolyticus* were detected in all conditions during the 7-day wet-storage test (Table 2). Depuration of lab-inoculated oysters in ASW for 48 h resulted in a large log reduction of *V. parahaemolyticus* in oysters from 2 to 4 log MPN/g, based on the temperature and salinity. The results indicated that *Vibrio* purging rates are higher during the first 2 days. Depuration efficiency at a salinity of 20 ppt was significantly different from that of the other two conditions. The result suggests that appropriate salinity of the ASW during wet-storage is necessary for depuration of the oysters since ours were harvested from aquaculture with a salinity of the water of around 10 ppt based on a previous study [2], a small reduction of *Vibrio* with a salinity of 10 is likely due to little or no biological activity of oysters occurring in seawater. A higher salinity (20 ppt) of wet-storage may favor the depuration of the oysters. However, the behavior of the oysters may be affected when we increase salinity to 25 ppt. Under this condition, the activity of the oysters may be inhibited; hence, *Vibrio* may not be depurated effectively. Chae et al. [17] reported that increased reductions of *V. parahaemolyticus* were observed when oysters were depurated at 15 °C. At low temperatures, the oyster pumps water at the highest level, hence the uptake of *Vibrio* would be retarded. The temperatures we choose for depuration in our study are all below 15 °C. Based on the results, we found that an optimum wet-storage process could be achieved at a combination of 7.5 °C temperature and 20 ppt salinity, which produced a 3.38 log (MPN/g) reduction after 7 days of depuration (Figure 1). The wet-storage process at 10 °C and salinity 20 ppt could also achieve a larger than 3 log reduction, which meets the FDA’s requirement of larger than a 3.0 log (MPN/g) reduction as a post-harvest process for oyster *V. parahaemolyticus* control [18]. We also found that the pre-chilling treatment could improve the efficacy of the wet-storage process.

Depuration has been treated as a post-harvest process for decreasing sewage-associated bacteria (such as coliforms, *Esherichia coli* and *Salmonella*). It has been widely applied in many countries to remove sewage-associated bacterial contaminants in different shellfish species sold alive [19,20]. However, based on the little effect on reducing *V. parahaemolyticus* in shellfish, depuration has not yet been practiced in the shellfish industry as a post-harvest treatment for controlling the natural flora of shellfish [14,21,22,23]. Several studies reported the efficacy of depuration in reducing *V. parahaemolyticus* in Pacific oysters [16,17,18,19]. One study detailed the efficacy of *V. parahaemolyticus* depuration in oysters [19]. However, there are several differences between our results and this study, especially in the effects of salinity on the first two days. In this study, the log reduction of *V. parahaemolyticus* in oysters after 1 day reached up to 2.83. In another study [2], their results show a 1.71 and 1.76 log reduction of *V. parahaemolyticus* in oysters after storage of 1 day at salinities of 20 and 25 ppt, respectively. In a third study [1], the log reduction of *V. parahaemolyticus* in oysters also reach up to 2.57 at a salinity of 30 ppt. Those results show better efficiency of depuration than our system. The reasons may be as follows: (1) oyster species—the previous studies used Pacific oysters while we used Eastern oysters for the experiments; (2) original environment—there could be differences in the growing water temperature and salinity ranges for the oysters in different studies, which may affect the activation of oysters during wet-storage; (3) upwelling design of the depuration systems—we notice that downwelling system was applied in previous studies, while in our study, we used an upwelling system. In a downwelling system, the purified water was introduced to the system from the upside and may avoid the resuspension of detritus. This may increase the efficiency of the depuration of the system. Future work will investigate in detail the effects of the design of the systems on depuration. We found many factors that affect the efficiency of wet-storage systems, other than on-site salinity, temperature, etc. In this study, we notice a significant difference in depuration efficiency between pre-chilled and non-pre-chilled tank water for *Vibrio* in oysters. Furthermore, we assume the source of the oysters (farm-raised or wild-caught) and accumulation method (laboratory inoculation or natural accumulation through temperature abuse) may affect the reduction of the *Vibrio* during wet-storage procedures. In addition, the shelf life of the oysters during and after depuration is another concern of the wet-storage process. Further large-scale research is needed to confirm these findings.

## 4. Materials and Methods

### 4.1. V. parahaemolyticus Cultures Preparation

Pathogenic (tlh+, tdh+, and trh+) and non-pathogenic (tlh+, tdh−, and trh−) *V. parahaemolyticus* strains isolated from clams and oysters were used in this study [24]. Each culture was individually grown in 10 mL tryptic soy broth (TSB, Difco, Becton) supplemented with 1% NaCl (TSB-Salt) at 37 °C for 18 to 24 h. The enriched cultures were streaked to individual tryptic soy agar (TSA, Difco, Becton Dickinson) supplemented with 1% NaCl (TSA-Salt) and incubated at 37 °C for 18 to 24 h. A single colony from a TSA-Salt plate was picked and enriched in 10 mL TSB-Salt at 37 °C for 4 h. The enriched cultures of *V. parahaemolyticus* were pooled into a 30 mL sterile centrifuge tube and centrifuged at 3000× *g* (To be checked) for 15 min. Pelleted cells were resuspended in 30 mL of sterile salt solution (1%) to produce a culture cocktail of approximately 4 × 10^8^ CFU/mL.

### 4.2. Oyster Preparation

Oysters were natural diploid adult eastern oysters sourced from an oyster farm in the Choptank River, MD. Oysters were transported on ice in a cooler to the lab within about 2 h. The oysters, after a brief washing with tap water to remove the mud on the shell, were ready to be inoculated with *Vibrio* spp. The ASW was prepared by dissolving Instant Ocean Salt (Instant Ocean, United Pet Group, Inc., Cincinnati, OH 45455, USA) in deionized water according to the manufacturer’s instructions.

### 4.3. Accumulation of V. parahaemolyticus in Oysters

Accumulation of *V. parahaemolyticus* in oysters was conducted according to previous studies [16,17]. For each depuration study, 150 aquaculture oysters or 75 commercial oysters were transferred to an identical HDPE tank containing 30 L of freshly prepared ASW containing a *V. parahaemolyticus* culture cocktail at a level of approximately 10^4^ to 10^5^ CFU/mL. The oysters were held in the tank at room temperature overnight (18 h). The salinity during the inoculation was controlled the same as that in the depuration process. Air was pumped into the water to keep dissolved oxygen (DO) levels favorable for oyster pumping and uptake of *Vibrio*. Levels of aquaculture oysters were analyzed with a 3-tube most probable-number (MPN) method before and after inoculation.

### 4.4. Oyster Depuration

After inoculation, the oysters were transferred to a laboratory-scale recirculating tank with 150 L of ASW. The system was equipped with an ultraviolet (UV) Lite (Emperor Aquatic smarts, Aqua Logic, Inc, San Diego, CA, USA), a water chiller (Delta Star, Aqua Logic Inc., San Diego, CA, USA), a thermometer, and a flow rate meter (1500 L/h). The temperature of the process was set at 7.5, 10, and 12.5 °C, while salinity was fixed at 15, 20, and 25 ppt. The levels of *V. parahaemolyticus* in oysters were analyzed every day for 7 days during the depuration process. A total of three studies (100 oysters per study) were conducted to assess the ability of oysters to survive in the refrigerated walk-in cooler (4 °C) after the accumulated treatments.

### 4.5. Microbiological Analysis

Levels of *V. parahaemolyticus* in oysters were analyzed with a 3-tube MPN method described in the U.S. Food and Drug Administration’s BAM [25], using thiosulfatecitrate-bile salts-sucrose agar (TCBS) and Chrom-agar for *V. parahaemolyticus* isolation and confirmation. The accuracy of using CHROM to determine the MPN values was confirmed by BAX PCR. In a preliminary study, we applied BAX PCR in a few cases to confirm isolates recovered from CHROM agar plates. We did not observe any significant differences between the two methods (using Chrom-agar or BAX PCR to confirm presumptive isolates taken in TCBS plates). Therefore, Chrom-agar was used in all tests of *V. parahaemolyticus* levels.

For each test, 6 oysters were shucked using a sterile knife on a sterile cutting board and blended in a sterile blender jar with an equal weight of sterile phosphate-buffered saline (PBS) at high speed for 90 s to produce a 1:1 shellfish diluent homogenate. To prepare a 1:10 dilution, 1 g of oyster homogenate (1:1) was mixed with 9 mL of PBS. Additional 10-fold dilutions were prepared using PBS (i.e., 1 mL of 1:10 to 9.0 mL of PBS for a 1:100 dilution). One-milliliter portions of all dilutions were individually inoculated into 3 tubes of 10 mL Alkaline Peptone Water (APW). APW tubes were incubated at 37 °C for 16–18 h and the positive (turbid) tubes were recorded. A loopful of each positive APW tube was streaked onto Thiosulfate-Citrate Bile Salt-Sucrose (TCBS) plates and incubated at 37 °C for 18–24 h. All plates that had green colonies were considered presumptive positive for *V. parahaemolyticus*. Then, three different suspicious colonies in each plate were looped to strike on Chromagar plates [26] and incubated at 37 °C for 18–24 h. All plates that had purple colonies were considered to be Vp positive.

### 4.6. Statistical Analysis

Results of microbiological tests were analyzed by using SPSS 19.0 software (Chicago, IL, USA). Significant differences among means of each treatment over time were established at a level of *p* < 0.05. All statistical analyses were based on 3 trials. Samples of each trial were collected from six oysters.

## 5. Conclusions

The wet-storage process at a combination of 7.5 °C temperature and 20 ppt salinity can achieve a 3.38 log (MPN/g) reduction at Day 7. A pre-chilled system, at the conditions of 10 °C and 20 ppt, can achieve a 3.54 log (MPN/g) reduction of *V. parahaemolyticus* in oysters at Day 7. There was no significant difference in the shelf life between inoculated and untreated oysters before the depuration, with the same survival rate (stored in a 4 °C cooler for 15 days) of 93%.

## Figures and Tables

**Figure 1 pathogens-11-00553-f001:**
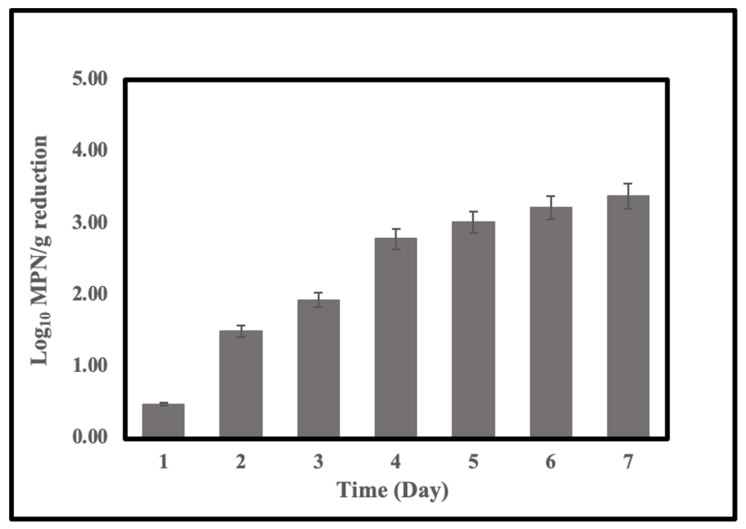
Reductions (Log_10_ MPN/g) of *V. parahaemolyticus* in lab-inoculated oysters during controlled re-circulating wet-storage process at 7.5 °C, salinity of 20 ppt. Data were collected as mean values of reductions from three trails (sample of each trail from six oysters).

**Figure 2 pathogens-11-00553-f002:**
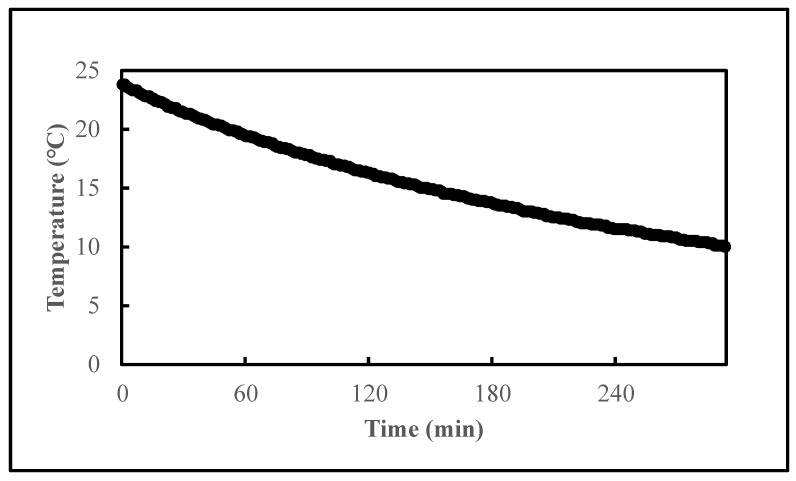
Temperature reduction curve of wet-storage system from room temperature to 10 °C.

**Figure 3 pathogens-11-00553-f003:**
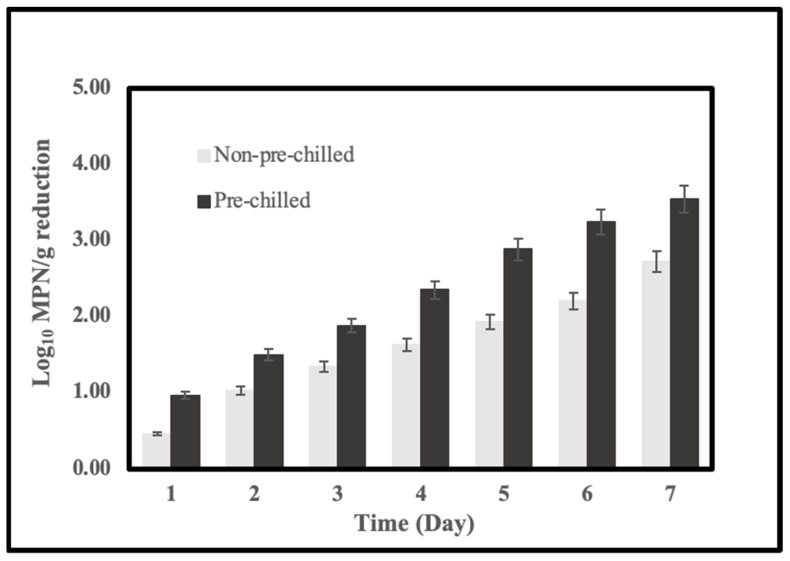
Reductions (Log10 MPN/g) of *V. parahaemolyticus* in lab-inoculated oysters during pre-chilled (black bar) and non-pre-chilled (grey bar) wet-storage processes at 10 °C, salinity of 20 ppt. Data were collected as mean values of reductions from three trails (sample of each trail from six oysters).

**Figure 4 pathogens-11-00553-f004:**
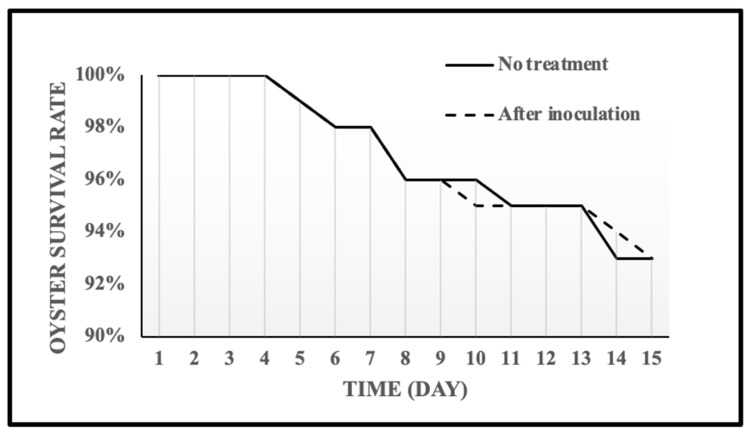
Survival of oysters after treatments for controlled re-circulating wet-storage process for 15 days.

**Table 1 pathogens-11-00553-t001:** *V. parahaemolyticus* levels (log_10_ MPN/g) in oysters at different dates of harvest.

*Vibrio* Levels	Date					
	28 November 2017	12 December 2017	10 April 2018	4 June 2018	2 July 2018	31 July 2018
Before inoculation	0.54 ± 0.16	0.45 ± 0.16	0.36 ± 0.00	2.44 ± 0.20	3.04 ± 0.15	3.31 ± 0.31
After inoculation	5.92 ± 0.22	5.41 ± 0.24	5.50 ± 0.28	5.79 ± 0.22	5.17 ± 0.21	5.79 ± 0.22

**Table 2 pathogens-11-00553-t002:** Changes of *V. parahaemolyticus* levels (log_10_ MPN/g) in lab-inoculated oysters during depuration at salinity of 15, 20, and 25 ppt. (Temperature was controlled at 10 °C).

Time (Day)	Salinity (ppt)		
	15	20	25
0	5.18 ± 0.00 ^A^	5.50 ± 0.28 ^A^	5.41 ± 0.24 ^A^
1	4.86 ± 0.19 ^B^	4.86 ± 0.19 ^B^	4.79 ± 0.22 ^B^
2	3.86 ± 0.19 ^C^	3.99 ± 0.37 ^C^	4.24 ± 0.12 ^C^
3	3.65 ± 0.30 ^CD^	3.75 ± 0.19 ^CD^	3.97 ± 0.00 ^CD^
4	3.30 ± 0.11 ^D^	3.30 ± 0.11 ^DE^	3.66 ± 0.00 ^DE^
5	2.97 ± 0.00 ^E^	2.97 ± 0.00 ^EF^	3.24 ± 0.24 ^EF^
6	2.53 ± 0.18 ^E^	2.63 ± 0.00 ^FG^	2.86 ± 0.19 ^FG^
7	2.32 ± 0.00 ^F^	2.36 ± 0.00 ^G^	2.63 ± 0.00 ^G^
Total log reduction	2.85 ± 0.00 ^a^	3.14 ± 0.00 ^b^	2.77 ± 0.12 ^a^
FDA Requirement Achieved (3 log reduction)	No	Yes	Yes

Data with the same letter in the same column are not significantly different (*p* > 0.05).

**Table 3 pathogens-11-00553-t003:** Changes of *V. parahaemolyticus* levels (log_10_ MPN/g) in laboratory-inoculated oysters during depuration at 7.5, 10, and 12.5 °C (Salinity was controlled at 20 ppt).

Time (Day)	Temperature (°C)		
	7.5	10	12.5
0	5.79 ± 0.22 ^A^	5.50 ± 0.28 ^A^	5.47 ± 0.16 ^A^
1	5.31 ± 0.12 ^B^	4.86 ± 0.19 ^B^	4.72 ± 0.29 ^B^
2	4.29 ± 0.11 ^C^	3.99 ± 0.37 ^C^	4.38 ± 0.00 ^BC^
3	3.86 ± 0.19 ^D^	3.75 ± 0.19 ^CD^	3.86 ± 0.28 ^CD^
4	3.01 ± 0.06 ^EF^	3.30 ± 0.11 ^DE^	3.65 ± 0.19 ^DE^
5	2.77 ± 0.12 ^F^	2.97 ± 0.00 ^EF^	3.57 ± 0.30 ^DEF^
6	2.57 ± 0.16 ^F^	2.63 ± 0.00 ^FG^	3.24 ± 0.16 ^EF^
7	2.41 ± 0.05 ^F^	2.36 ± 0.00 ^G^	3.04 ± 0.12 ^F^
Total log reduction	3.38 ± 0.05 ^a^	3.14 ± 0.00 ^b^	2.44 ± 0.12 ^c^
FDA Requirement Achieved (3 log reduction)	Yes	Yes	No

Data with the same letter in the same column are not significantly different (*p* > 0.05).

## Data Availability

Not applicable.

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
