# Peer review of "Controlled Recirculating Wet Storage Purging V. parahaemolyticus in Oysters"

_pathogens, 2022, doi:10.3390/pathogens11050553_

Round 1
Reviewer 1 Report
While the paper is generally easy to read, some general grammatical issues need to be addressed. Results should be presented in past tense. There are multiple errors with subject/verb agreement. This should be checked carefully.
Throughout, the authors refer to V. parahaemolyticus "populations". It would be more accurate to state "levels" or "abundances" as the data is documenting changes in values of added (for the most part) vibrio, not natural populations. Also, only targeting detection of Vp doesn't provide information about the "population" other that the total amount present.
L44, irradiation also reduces vibrio while keeping the oysters alive
L62, I think you mean 3 log MPN/g?
Table 1. Clarify what the two lines of the table represent. It appears 1 is natural levels and 2 is levels post-inoculation, but this is not clearly stated.
Tables (or associated text) clarify what the +/- is (SD?) and how many samples are represented by each value.
L87, italicize V. parahaemolyticus
Figure 1. Explain what the error bars represent. I am not a modeler, but a linear regression does not seem to be the most appropriate fit for these data. There is a clear flattening of reduction starting at day 4, so following the linear prediction would be inappropriate. Please change.
Section 2.4, why wasn't the pre-chilling tested with "optimal" conditions of 7.5C and 20ppt? Please explain.
The image for Figure 3 does not match the legend. It appears to be a duplicate of Figure 4.
L118-9, it is unclear why a temp abuse treatment was included in the shelf life study. Was a control group included ("normal" handling)? What about oysters that had gone through the depuration process? That is the most important group to examine in relation to this study. That data should be provided.
L152, "uptake of vibrio" is more accurate than "growth". Please modify.
L154-5, please clarify why this condition is listed as optimal when the pre-chilling provided a greater log reduction.
L220, please clarify if the temp abuse was a separate method or used following inoculation to further increase the vibrio load.
L228, please provide the flow rate data for these experiments.
Section 4.4, how many times were each of the depuration conditions tested? This information is critical to determining the ruggedness of the study.
L237-8, here describes 6 oysters per sample, but in the figure legends above, it was stated that the means were of 5 individual oysters. Please clarify.
L247-254, please clarify this section. It sounds like suspect Vp from TCBS were streaked to CHROM and if also suspect on CHROM confirmed using BAX. Then the BAX results used to determine MPN values. But, as written it is unclear.
Reviewer 2 Report
I would like the novel aspect of this work to be clearly indicated and justified. Similar studies (Effects of Temperature and Salinity on Vibrio vulnificus Population Dynamics as Assessed by Quantitative PCR; Mark A. Randa et al., 2004) have been carried out many years ago, so the reviewer does not find the novelty of this study carried out after 18 years.
There is no highlight in this paper.
indicate microorganisms in italics
Introduction
- A review of the references is required to update them. An example would be 7 and 8, which are very old data.
- Some previous reference is required to justify the pre-chilled treatment
Methods
- Indicate the origin of the strains. If they are from a collection, indicate the reference.
- The author indicates that the strains are isolates. Indicate procedure to isolate these bacteria.
- indicate the reference of the Chromagar plates
- Some previous reference is required to justify the pre-chilled treatment
- Explain the method used (the 3-tube MPN calculator)
Results
- qPCR results are required for Concentrations of Vp
Conclusions
- Vibrio or Vibrio parahaemolyticus
Bibliography format is not uniform
Round 2
Reviewer 2 Report
This version is correct for publication